# Distributed Submodular Maximization: Identifying Representative Elements in Massive Data

**Baharan Mirzasoleiman**
ETH Zurich

**Amin Karbasi**
ETH Zurich

**Rik Sarkar**
University of Edinburgh

**Andreas Krause**
ETH Zurich

## Abstract

Many large-scale machine learning problems (such as clustering, non-parametric learning, kernel machines, etc.) require selecting, out of a massive data set, a manageable yet representative subset. Such problems can often be reduced to maximizing a submodular set function subject to cardinality constraints. Classical approaches require centralized access to the full data set; but for truly large-scale problems, rendering the data centrally is often impractical. In this paper, we consider the problem of submodular function maximization in a distributed fashion. We develop a simple, two-stage protocol GREEDI, that is easily implemented using MapReduce style computations. We theoretically analyze our approach, and show, that under certain natural conditions, performance close to the (impractical) centralized approach can be achieved. In our extensive experiments, we demonstrate the effectiveness of our approach on several applications, including sparse Gaussian process inference and exemplar-based clustering, on tens of millions of data points using Hadoop.

## 1 Introduction

Numerous machine learning algorithms require selecting representative subsets of manageable size out of large data sets. Applications range from exemplar-based clustering [1], to active set selection for large-scale kernel machines [2], to corpus subset selection for the purpose of training complex prediction models [3]. Many such problems can be reduced to the problem of maximizing a submodular set function subject to cardinality constraints [4, 5].

Submodularity is a property of set functions with deep theoretical and practical consequences. Submodular maximization generalizes many well-known problems, e.g., maximum weighted matching, max coverage, and finds numerous applications in machine learning and social networks, such as influence maximization [6], information gathering [7], document summarization [3] and active learning [8, 9]. A seminal result of Nemhauser et al. [10] states that a simple greedy algorithm produces solutions competitive with the optimal (intractable) solution. In fact, if assuming nothing but submodularity, no efficient algorithm produces better solutions in general [11, 12].

Data volumes are increasing faster than the ability of individual computers to process them. Distributed and parallel processing is therefore necessary to keep up with modern massive datasets. The greedy algorithms that work well for centralized submodular optimization, however, are unfortunately sequential in nature; therefore they are poorly suited for parallel architectures. This mismatch makes it inefficient to apply classical algorithms directly to distributed setups.

In this paper, we develop a simple, parallel protocol called GREEDI for distributed submodular maximization. It requires minimal communication, and can be easily implemented in MapReduce style parallel computation models [13]. We theoretically characterize its performance, and show that under some natural conditions, for large data sets the quality of the obtained solution is competitive with the best centralized solution. Our experimental results demonstrate the effectiveness of our approach on a variety of submodular maximization problems. We show that for problems such as exemplar-based clustering and active set selection, our approach leads to parallel solutions that are very competitive with those obtained via centralized methods ($98\%$ in exemplar based clustering and $97\%$ in active set selection). We implement our approach in Hadoop, and show how it enables sparse Gaussian process inference and exemplar-based clustering on data sets containing tens of millions of points.

## 2   Background and Related Work

Due to the rapid increase in data set sizes, and the relatively slow advances in sequential processing capabilities of modern CPUs, parallel computing paradigms have received much interest. Inhabiting a sweet spot of resiliency, expressivity and programming ease, the MapReduce style computing model [13] has emerged as prominent foundation for large scale machine learning and data mining algorithms [14, 15]. MapReduce works by distributing the data to independent machines, where it is processed in parallel by *map tasks* that produce key-value pairs. The output is shuffled, and combined by *reduce* tasks. Hereby, each *reduce* task processes inputs that share the same key. Their output either comprises the ultimate result, or forms the input to another MapReduce computation.

The problem of centralized maximization of submodular functions has received much interest, starting with the seminal work of [10]. Recent work has focused on providing approximation guarantees for more complex constraints. See [5] for a recent survey. The work in [16] considers an algorithm for online distributed submodular maximization with an application to sensor selection. However, their approach requires $k$ stages of communication, which is unrealistic for large $k$ in a MapReduce style model. The authors in [4] consider the problem of submodular maximization in a streaming model; however, their approach is not applicable to the general distributed setting. There has also been new improvements in the running time of the greedy solution for solving SET-COVER when the data is large and disk resident [17]. However, this approach is not parallelizable by nature.

Recently, specific instances of distributed submodular maximization have been studied. Such scenarios often occur in large-scale graph mining problems where the data itself is too large to be stored on one machine. Chierichetti et al. [18] address the MAX-COVER problem and provide a $(1-1/e-\epsilon)$ approximation to the centralized algorithm, however at the cost of passing over the data set many times. Their result is further improved by Blelloch et al. [19]. Lattanzi et al. [20] address more general graph problems by introducing the idea of filtering, namely, reducing the size of the input in a distributed fashion so that the resulting, much smaller, problem instance can be solved on a single machine. This idea is, in spirit, similar to our distributed method GREEDI. In contrast, we provide a more general framework, and analyze in which settings performance competitive with the centralized setting can be obtained.

## 3   The Distributed Submodular Maximization Problem

We consider the problem of selecting subsets out of a large data set, indexed by $V$ (called ground set). Our goal is to maximize a non-negative set function $f : 2^V \rightarrow \mathbb{R}_+$, where, for $S \subseteq V$, $f(S)$ quantifies the utility of set $S$, capturing, e.g., how well $S$ represents $V$ according to some objective. We will discuss concrete instances of functions $f$ in Section 3.1. A set function $f$ is naturally associated with a *discrete derivative*

$$\triangle_f(e|S) \doteq f(S \cup \{e\}) - f(S), \tag{1}$$

where $S \subseteq V$ and $e \in V$, which quantifies the increase in utility obtained when adding $e$ to set $S$. $f$ is called *monotone* iff for all $e$ and $S$ it holds that $\triangle_f(e|S) \geq 0$. Further, $f$ is *submodular* iff for all $A \subseteq B \subseteq V$ and $e \in V \setminus B$ the following diminishing returns condition holds:

$$\triangle_f(e|A) \geq \triangle_f(e|B). \tag{2}$$

Throughout this paper, we focus on such monotone submodular functions. For now, we adopt the common assumption that $f$ is given in terms of a value oracle (a black box) that computes $f(S)$ for any $S \subseteq V$. In Section 4.5, we will discuss the setting where $f(S)$ itself depends on the entire data set $V$, and not just the selected subset $S$. Submodular functions contain a large class of functions that naturally arise in machine learning applications (c.f., [5, 4]). The simplest example of such functions are *modular* functions for which the inequality (2) holds with equality.

The focus of this paper is on maximizing a monotone submodular function (subject to some constraint) in a distributed manner. Arguably, the simplest form of constraints are cardinality constraints. More precisely, we are interested in the following optimization problem:

$$\max_{S \subseteq V} f(S) \quad \text{s.t.} \quad |S| \leq k. \tag{3}$$

We will denote by $A^{\mathrm{c}}[k]$ the subset of size at most $k$ that achieves the above maximization, i.e., the best centralized solution. Unfortunately, problem (3) is NP-hard, for many classes of submodular functions [12]. However, a seminal result by Nemhauser et al. [10] shows that a simple greedy algorithm provides a $(1 - 1/e)$ approximation to (3). This greedy algorithm starts with the empty set $S_0$, and at each iteration $i$, it chooses an element $e \in V$ that maximizes (1), i.e., $S_i = S_{i-1} \cup \{\arg\max_{e \in V} \triangle_f(e|S_{i-1})\}$. Let $A^{\mathrm{gc}}[k]$ denote this greedy-centralized solution of size at most $k$. For several classes of monotone submodular functions, it is known that $(1 - 1/e)$ is the best approximation guarantee that one can hope for [11, 12, 21]. Moreover, the greedy algorithm can be accelerated using lazy evaluations [22].

In many machine learning applications where the ground set $|V|$ is large (e.g., cannot be stored on a single computer), running a standard greedy algorithm or its variants (e.g., lazy evaluation) in a centralized manner is infeasible. Hence, in those applications we seek a distributed solution, e.g., one that can be implemented using MapReduce-style computations (see Section 5). From the algorithmic point of view, however, the above greedy method is in general difficult to parallelize, since at each step, only the object with the highest marginal gain is chosen and every subsequent step depends on the preceding ones. More precisely, the problem we are facing in this paper is the following. Let the ground set $V$ be partitioned into $V_1, V_2, \ldots, V_m$, i.e., $V = V_1 \cup V_2, \cdots \cup V_m$ and $V_i \cap V_j = \emptyset$ for $i \neq j$. We can think of $V_i$ as a subset of elements (e.g., images) on machine $i$. The questions we are trying to answer in this paper are: how to distribute $V$ among $m$ machines, which algorithm should run on each machine, and how to merge the resulting solutions.

### 3.1 Example Applications Suitable for Distributed Submodular Maximization

In this part, we discuss two concrete problem instances, with their corresponding submodular objective functions $f$, where the size of the datasets often requires a distributed solution for the underlying submodular maximization.

**Active Set Selection in Sparse Gaussian Processes (GPs):** Formally a GP is a joint probability distribution over a (possibly infinite) set of random variables $\mathbf{X}_V$, indexed by our ground set $V$, such that every (finite) subset $\mathbf{X}_S$ for $S = \{e_1, \ldots, e_s\}$ is distributed according to a multivariate normal distribution, i.e., $P(\mathbf{X}_S = \mathbf{x}_S) = \mathcal{N}(\mathbf{x}_S; \mu_S, \Sigma_{S,S})$, where $\mu_S = (\mu_{e_1}, \ldots, \mu_{e_s})$ and $\Sigma_{S,S} = [\mathcal{K}_{e_i, e_j}](1 \leq i, j \leq k)$ are the prior mean vector and prior covariance matrix, respectively. The covariance matrix is parametrized via a (positive definite kernel) function $\mathcal{K}$. For example, a commonly used kernel function in practice where elements of the ground set $V$ are embedded in a Euclidean space is the squared exponential kernel $\mathcal{K}_{e_i, e_j} = \exp(-|e_i - e_j|_2^2/h^2)$. In GP regression, each data point $e \in V$ is considered a random variable. Upon observations $\mathbf{y}_A = \mathbf{x}_A + \mathbf{n}_A$ (where $\mathbf{n}_A$ is a vector of independent Gaussian noise with variance $\sigma^2$), the predictive distribution of a new data point $e \in V$ is a normal distribution $P(\mathbf{X}_e \mid \mathbf{y}_A) = \mathcal{N}(\mu_{e|A}, \Sigma_{e|A}^2)$, where

$$\mu_{e|A} = \mu_e + \Sigma_{e,A}(\Sigma_{A,A} + \sigma^2 \mathbf{I})^{-1}(\mathbf{x}_A - \mu_A), \quad \sigma_{e|A}^2 = \sigma_e^2 - \Sigma_{e,A}(\Sigma_{A,A} + \sigma^2 \mathbf{I})^{-1}\Sigma_{A,e}. \tag{4}$$

Note that evaluating (4) is computationally expensive as it requires a matrix inversion. Instead, most efficient approaches for making predictions in GPs rely on choosing a small – so called *active* – set of data points. For instance, in the Informative Vector Machine (IVM) one seeks a set $S$ such that the information gain, $f(S) = I(\mathbf{Y}_S; \mathbf{X}_V) = H(\mathbf{X}_V) - H(\mathbf{X}_V | \mathbf{Y}_S) = \frac{1}{2} \log \det(\mathbf{I} + \sigma^{-2}\Sigma_{S,S})$ is maximized. It can be shown that this choice of $f$ is monotone submodular [21]. For medium-scale problems, the standard greedy algorithms provide good solutions. In Section 5, we will show how GREEDI can choose near-optimal subsets out of a data set of 45 million vectors.

**Exemplar Based Clustering:**   Suppose we wish to select a set of exemplars, that best represent a massive data set. One approach for finding such exemplars is solving the $k$-medoid problem [23], which aims to minimize the sum of pairwise dissimilarities between exemplars and elements of the dataset. More precisely, let us assume that for the data set $V$ we are given a distance function $d : V \times V \to \mathbb{R}$ (not necessarily assumed symmetric, nor obeying the triangle inequality) such that $d(\cdot, \cdot)$ encodes dissimilarity between elements of the underlying set $V$. Then, the loss function for $k$-medoid can be defined as follows: $L(S) = \frac{1}{|V|} \sum_{e \in V} \min_{v \in S} d(e, v)$. By introducing an auxiliary element $e_0$ (e.g., $= 0$) we can turn $L$ into a monotone submodular function: $f(S) = L(\{e_0\}) - L(S \cup \{e_0\})$. In words, $f$ measures the decrease in the loss associated with the set $S$ versus the loss associated with just the auxiliary element. It is easy to see that for suitable choice of $e_0$, maximizing $f$ is equivalent to minimizing $L$. Hence, the standard greedy algorithm provides a very good solution. But again, the problem becomes computationally challenging when we have a large data set and we wish to extract a small set of exemplars. Our distributed solution GREEDI addresses this challenge.

### 3.2   Naive Approaches Towards Distributed Submodular Maximization

One way of implementing the greedy algorithm in parallel would be the following. We proceed in rounds. In each round, all machines – in parallel – compute the marginal gains of all elements in their sets $V_i$. They then communicate their candidate to a central processor, who identifies the globally best element, which is in turn communicated to the $m$ machines. This element is then taken into account when selecting the next element and so on. Unfortunately, this approach requires synchronization after each of the $k$ rounds. In many applications, $k$ is quite large (e.g., tens of thousands or more), rendering this approach impractical for MapReduce style computations.

An alternative approach for large $k$ would be to – on each machine – greedily select $k/m$ elements independently (without synchronization), and then merge them to obtain a solution of size $k$. This approach is much more communication efficient, and can be easily implemented, e.g., using a single MapReduce stage. Unfortunately, many machines may select redundant elements, and the merged solution may suffer from diminishing returns.

In Section 4, we introduce an alternative protocol GREEDI, which requires little communication, while at the same time yielding a solution competitive with the centralized one, under certain natural additional assumptions.

## 4   The GREEDI Approach for Distributed Submodular Maximization

In this section we present our main results. We first provide our distributed solution GREEDI for maximizing submodular functions under cardinality constraints. We then show how we can make use of the geometry of data inherent in many practical settings in order to obtain strong data-dependent bounds on the performance of our distributed algorithm.

### 4.1   An Intractable, yet Communication Efficient Approach

Before we introduce GREEDI, we first consider an intractable, but communication–efficient parallel protocol to illustrate the ideas. This approach, shown in Alg. 1, first distributes the ground set $V$ to $m$ machines. Each machine then finds the *optimal* solution, i.e., a set of cardinality at most $k$, that maximizes the value of $f$ in each partition. These solutions are then merged, and the optimal subset of cardinality $k$ is found in the combined set. We call this solution $f(A^{\mathrm{d}}[m, k])$.

As the optimum centralized solution $A^{\mathrm{c}}[k]$ achieves the maximum value of the submodular function, it is clear that $f(A^{\mathrm{c}}[k]) \geq f(A^{\mathrm{d}}[m, k])$. Further, for the special case of selecting a single element $k = 1$, we have $A^{\mathrm{c}}[1] = A^{\mathrm{d}}[m, 1]$. In general, however, there is a gap between the distributed and the centralized solution. Nonetheless, as the following theorem shows, this gap cannot be more than $1/\min(m, k)$. Furthermore, this is the best result one can hope for under our two-round model.

**Theorem 4.1.** *Let $f$ be a monotone submodular function and let $k > 0$. Then, $f(A^{\mathrm{d}}[m, k]) \geq \frac{1}{\min(m,k)} f(A^{\mathrm{c}}[k])$. In contrast, for any value of $m$, and $k$, there is a data partition and a monotone submodular function $f$ such that $f(A^{\mathrm{c}}[k]) = \min(m, k) \cdot f(A^{\mathrm{d}}[m, k])$.*

| **Algorithm 1** Exact Distrib. Submodular Max. | **Algorithm 2** Greedy Dist. Subm. Max. (GREEDI) |
|---|---|
| **Input:** Set $V$, #of partitions $m$, constraints $k$. | **Input:** Set $V$, #of partitions $m$, constraints $l$, $\kappa$. |
| **Output:** Set $A^{\mathrm{d}}[m,k]$. | **Output:** Set $A^{\mathrm{gd}}[m,\kappa,l]$. |
| 1: Partition $V$ into $m$ sets $V_1, V_2, \ldots, V_m$. | 1: Partition $V$ into $m$ sets $V_1, V_2, \ldots, V_m$. |
| 2: In each partition $V_i$ find the optimum set $A_i^c[k]$ of cardinality $k$. | 2: Run the standard greedy algorithm on each set $V_i$. Find a solution $A_i^{gc}[\kappa]$. |
| 3: Merge the resulting sets: $B = \cup_{i=1}^m A_i^c[k]$. | 3: Merge the resulting sets: $B = \cup_{i=1}^m A_i^{gc}[\kappa]$. |
| 4: Find the optimum set of cardinality $k$ in $B$. Output this solution $A^{\mathrm{d}}[m,k]$. | 4: Run the standard greedy algorithm on $B$ until $l$ elements are selected. Return $A^{\mathrm{gd}}[m,\kappa,l]$. |

The proof of all the theorems can be found in the supplement. The above theorem fully characterizes the performance of two-round distributed algorithms in terms of the best centralized solution. A similar result in fact also holds for non-negative (not necessarily monotone) functions. Due to space limitation, the result is reported in the appendix. In practice, we cannot run Alg. 1. In particular, there is no efficient way to identify the optimum subset $A_i^c[k]$ in set $V_i$, unless P=NP. In the following, we introduce our efficient approximation GREEDI.

## 4.2 Our GREEDI Approximation

Our main efficient distributed method GREEDI is shown in Algorithm 2. It parallels the intractable Algorithm 1, but replaces the selection of optimal subsets by a greedy algorithm. Due to the approximate nature of the greedy algorithm, we allow the algorithms to pick sets slightly larger than $k$. In particular, GREEDI is a two-round algorithm that takes the ground set $V$, the number of partitions $m$, and the cardinality constraints $l$ (final solution) and $\kappa$ (intermediate outputs). It first distributes the ground set over $m$ machines. Then each machine separately runs the standard greedy algorithm, namely, it sequentially finds an element $e \in V_i$ that maximizes the discrete derivative shown in (1). Each machine $i$ – in parallel – continues adding elements to the set $A_i^{gc}[\cdot]$ until it reaches $\kappa$ elements. Then the solutions are merged: $B = \cup_{i=1}^m A_i^{gc}[\kappa]$, and another round of greedy selection is performed over $B$, which this time selects $l$ elements. We denote this solution by $A^{\mathrm{gd}}[m,\kappa,l]$: the greedy solution for parameters $m, \kappa$ and $l$. The following result parallels Theorem 4.1.

**Theorem 4.2.** *Let $f$ be a monotone submodular function and let $l, \kappa, k > 0$. Then*

$$f(A^{gd}[m,\kappa,l])) \geq \frac{(1 - e^{-\kappa/k})(1 - e^{-l/\kappa})}{\min(m,k)} f(A^c[k]).$$

For the special case of $\kappa = l = k$ the result of 4.2 simplifies to $f(A^{\mathrm{gd}}[m,\kappa,k]) \geq \frac{(1-1/e)^2}{\min(m,k)} f(A^c[k])$. From Theorem 4.1, it is clear that in general one cannot hope to eliminate the dependency of the distributed solution on $\min(k,m)$. However, as we show below, in many practical settings, the ground set $V$ and $f$ exhibit rich geometrical structure that can be used to prove stronger results.

## 4.3 Performance on Datasets with Geometric Structure

In practice, we can hope to do much better than the worst case bounds shown above by exploiting underlying structures often present in real data and important set functions. In this part, we assume that a metric $d$ exists on the data elements, and analyze performance of the algorithm on functions that change gracefully with change in the input. We refer to these as *Lipschitz functions.* More formally, a function $f : 2^V \to \mathbb{R}$ is $\lambda$-*Lipschitz*, if for equal sized sets $S = \{e_1, e_2, \ldots, e_k\}$ and $S' = \{e_1', e_2', \ldots, e_k'\}$ and for any matching of elements: $M = \{(e_1, e_1'), (e_2, e_2') \ldots, (e_k, e_k')\}$, the difference between $f(S)$ and $f(S')$ is bounded by the total of distances between respective elements: $|f(S) - f(S')| \leq \lambda \sum_i d(e_i, e_i')$. It is easy to see that the objective functions from both examples in Section 3.1 are $\lambda$-Lipschitz for suitable kernels/distance functions. Two sets $S$ and $S'$ are $\varepsilon$-*close* with respect to $f$, if $|f(S) - f(S')| \leq \varepsilon$. Sets that are close with respect to $f$ can be thought as good candidates to approximate the value of $f$ over each-other; thus one such set is a good representative of the other. Our goal is to find sets that are suitably close to $A^c[k]$. At an element $v \in V$, let us define its $\alpha$-*neighborhood* to be the set of elements within a distance $\alpha$ from

$v$ (i.e., $\alpha$-close to $v$): $N_\alpha(v) = \{w : d(v,w) \leq \alpha\}$. We can in general consider $\alpha$-neighborhoods of points of the metric space.

Our algorithm GREEDI partitions $V$ into sets $V_1, V_2, \ldots V_m$ for parallel processing. In this subsection, *we assume that* GREEDI *performs the partition by assigning elements uniformly randomly to the machines.* The following theorem says that if the $\alpha$-neighborhoods are sufficiently dense and $f$ is a $\lambda$-lipschitz function, then this method can produce a solution close to the centralized solution:

**Theorem 4.3.** *If for each* $e_i \in A^c[k], |N_\alpha(e_i)| \geq km \log(k/\delta^{1/m})$, *and algorithm* GREEDI *assigns elements uniformly randomly to $m$ processors , then with probability at least* $(1 - \delta)$,

$$f(A^{gd}[m,\kappa,l]) \geq (1 - e^{-\kappa/k})(1 - e^{-l/\kappa})(f(A^c[k]) - \lambda\alpha k).$$

### 4.4 Performance Guarantees for Very Large Data Sets

Suppose that our data set is a finite sample drawn from an underlying *infinite* set, according to some unknown probability distribution. Let $A^c[k]$ be an optimal solution in the infinite set such that around each $e_i \in A^c[k]$, there is a neighborhood of radius at least $\alpha^*$, where the probability density is at least $\beta$ at all points, for some constants $\alpha^*$ and $\beta$. This implies that the solution consists of elements coming from reasonably dense and therefore representative regions of the data set.

Let us consider $g : \mathbb{R} \to \mathbb{R}$, the *growth function of the metric*. $g(\alpha)$ is defined to be the volume of a ball of radius $\alpha$ centered at a point in the metric space. This means, for $e_i \in A^c[k]$ the probability of a random element being in $N_\alpha(e_i)$ is at least $\beta g(\alpha)$ and the expected number of $\alpha$ neighbors of $e_i$ is at least $E[|N_\alpha(e_i)|] = n\beta g(\alpha)$. As a concrete example, Euclidean metrics of dimension $D$ have $g(\alpha) = O(\alpha^D)$. Note that for simplicity we are assuming the metric to be homogeneous, so that the growth function is the same at every point. For heterogeneous spaces, we require $g$ to be a uniform lower bound on the growth function at every point.

In these circumstances, the following theorem guarantees that if the data set $V$ is sufficiently large and $f$ is a $\lambda$-lipschitz function, then GREEDI produces a solution close to the centralized solution.

**Theorem 4.4.** *For* $n \geq \dfrac{8km \log(k/\delta^{1/m})}{\beta g(\frac{\varepsilon}{\lambda k})}$, *where* $\frac{\varepsilon}{\lambda k} \leq \alpha^*$, *if the algorithm* GREEDI *assigns elements uniformly randomly to $m$ processors , then with probability at least* $(1 - \delta)$,

$$f(A^{gd}[m,\kappa,l]) \geq (1 - e^{-\kappa/k})(1 - e^{-l/\kappa})(f(A^c[k]) - \varepsilon).$$

### 4.5 Handling Decomposable Functions

So far, we have assumed that the objective function $f$ is given to us as a black box, which we can evaluate for any given set $S$ *independently* of the data set $V$. In many settings, however, the objective $f$ depends itself on the entire data set. In such a setting, we cannot use GREEDI as presented above, since we cannot evaluate $f$ on the individual machines without access to the full set $V$. Fortunately, many such functions have a simple structure which we call *decomposable*. More precisely, we call a monotone submodular function $f$ *decomposable* if it can be written as a sum of (non-negative) monotone submodular functions as follows: $f(S) = \frac{1}{|V|}\sum_{i \in V} f_i(S)$. In other words, there is separate monotone submodular function associated with every data point $i \in V$. We require that each $f_i$ can be evaluated without access to the full set $V$. Note that the exemplar based clustering application we discussed in Section 3.1 is an instance of this framework, among many others.

Let us define the evaluation of $f$ restricted to $D \subseteq V$ as follows: $f_D(S) = \frac{1}{|D|}\sum_{i \in D} f_i(S)$. Then, in the remaining of this section, our goal is to show that assigning each element of the data set randomly to a machine and running GREEDI will provide a solution that is with high probability close to the optimum solution. For this, let us assume the $f_i$'s are bounded, and without loss of generality $0 \leq f_i(S) \leq 1$ for $1 \leq i \leq |V|, S \subseteq V$. Similar to Section 4.3 we assume that GREEDI performs the partition by assigning elements uniformly randomly to the machines. These machines then each greedily optimize $f_{V_i}$. The second stage of GREEDI optimizes $f_U$, where $U \subseteq V$ is chosen uniformly at random, of size $\lceil n/m \rceil$. Then, we can show the following result.

**Theorem 4.5.** *Let* $m, k, \delta > 0$, $\epsilon < 1/4$ *and let* $n_0$ *be an integer such that for* $n \geq n_0$ *we have* $\ln(n)/n \leq \epsilon^2/(mk)$. *For* $n \geq \max(n_0, m\log(\delta/4m)/\epsilon^2)$, *and under the assumptions of Theorem 4.4, we have, with probability at least* $1 - \delta$,

$$f(A^{gd}[m, \kappa, l]) \geq (1 - e^{-\kappa/k})(1 - e^{-l/\kappa})(f(A^c[k]) - 2\varepsilon).$$

The above result demonstrates why GREEDI performs well on decomposable submodular functions with massive data even when they are evaluated locally on each machine. We will report our experimental results on exemplar-based clustering in the next section.

## 5  Experiments

In our experimental evaluation we wish to address the following questions: 1) how well does GREEDI perform compared to a centralized solution, 2) how good is the performance of GREEDI when using decomposable objective functions (see Section 4.5), and finally 3) how well does GREEDI scale on massive data sets. To this end, we run GREEDI on two scenarios: exemplar based clustering and active set selection in GPs. Further experiments are reported in the supplement.

We compare the performance of our GREEDI method (using different values of $\alpha = \kappa/k$) to the following naive approaches: a) *random/random*: in the first round each machine simply outputs $k$ randomly chosen elements from its local data points and in the second round $k$ out of the merged $mk$ elements, are again randomly chosen as the final output. b) *random/greedy*: each machine outputs $k$ randomly chosen elements from its local data points, then the standard greedy algorithm is run over $mk$ elements to find a solution of size $k$. c) *greedy/merge*: in the first round $k/m$ elements are chosen greedily from each machine and in the second round they are merged to output a solution of size $k$. d) *greedy/max*: in the first round each machine greedily finds a solution of size $k$ and in the second round the solution with the maximum value is reported. For data sets where we are able to find the centralized solution, we report the ratio of $f(A_{\text{dist}}[k])/f(A^{gc}[k])$, where $A_{\text{dist}}[k]$ is the distributed solution (in particular $A^{gd}[m, \alpha k, k] = A_{\text{dist}}[k]$ for GREEDI).

**Exemplar based clustering.** Our exemplar based clustering experiment involves GREEDI applied to the clustering utility $f(S)$ (see Sec. 3.1) with $d(x, x') = \|x - x'\|^2$. We performed our experiments on a set of 10,000 *Tiny Images* [24]. Each 32 by 32 RGB pixel image was represented by a 3,072 dimensional vector. We subtracted from each vector the mean value, normalized it to unit norm, and used the origin as the auxiliary exemplar. Fig. 1a compares the performance of our approach to the benchmarks with the number of exemplars set to $k = 50$, and varying number of partitions $m$. It can be seen that GREEDI significantly outperforms the benchmarks and provides a solution that is very close to the centralized one. Interestingly, even for very small $\alpha = \kappa/k < 1$, GREEDI performs very well. Since the exemplar based clustering utility function is decomposable, we repeated the experiment for the more realistic case where the function evaluation in each machine was restricted to the local elements of the dataset in that particular machine (rather than the entire dataset). Fig 1b shows similar qualitative behavior for decomposable objective functions.

***Large scale experiments with Hadoop.*** As our first large scale experiment, we applied GREEDI to the whole dataset of 80,000,000 *Tiny Images* [24] in order to select a set of 64 exemplars. Our experimental infrastructure was a cluster of 10 quad-core machines running Hadoop with the number of reducers set to $m = 8000$. Hereby, each machine carried out a set of reduce tasks in sequence. We first partitioned the images uniformly at random to reducers. Each reducer separately performed the lazy greedy algorithm on its own set of 10,000 images ($\approx$123MB) to extract 64 images with the highest marginal gains w.r.t. the local elements of the dataset in that particular partition. We then merged the results and performed another round of lazy greedy selection on the merged results to extract the final 64 exemplars. Function evaluation in the second stage was performed w.r.t a randomly selected subset of 10,000 images from the entire dataset. The maximum running time per reduce task was 2.5 hours. As Fig. 1c shows, GREEDI highly outperforms the other distributed benchmarks and can scale well to very large datasets. Fig. 1d shows a set of cluster exemplars discovered by GREEDI where each column in Fig. 1h shows 8 nearest images to exemplars 9 and 16 (shown with red borders) in Fig. 1d.

**Active set selection.** Our active set selection experiment involves GREEDI applied to the information gain $f(S)$ (see Sec. 3.1) with Gaussian kernel, $h = 0.75$ and $\sigma = 1$. We used the *Parkinsons Telemonitoring* dataset [25] consisting of 5,875 bio-medical voice measurements with 22 attributes

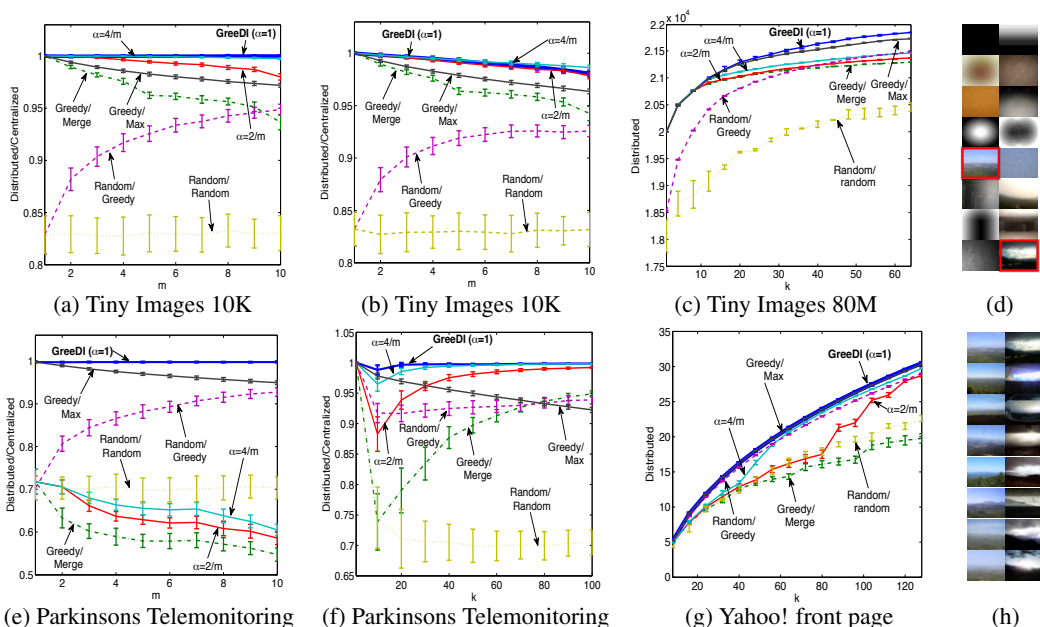

(a) Tiny Images 10K    (b) Tiny Images 10K    (c) Tiny Images 80M    (d)

(e) Parkinsons Telemonitoring    (f) Parkinsons Telemonitoring    (g) Yahoo! front page    (h)

Figure 1: Performance of GREEDI compared to the other benchmarks. a) and b) show the mean and standard deviation of the ratio of distributed vs. centralized solution for global and local objective functions with budget $k = 50$ and varying the number $m$ of partitions, for a set of 10,000 *Tiny Images*. c) shows the distributed solution with $m = 8000$ and varying $k$ for local objective functions on the whole dataset of 80,000,000 *Tiny Images*. e) shows the ratio of distributed vs. centralized solution with $m = 10$ and varying $k$ for *Parkinsons Telemonitoring*. f) shows the same ratio with $k = 50$ and varying $m$ on the same dataset, and g) shows the distributed solution for $m = 32$ with varying budget $k$ on *Yahoo! Webscope data*. d) shows a set of cluster exemplars discovered by GREEDI, and each column in h) shows 8 images nearest to exemplars 9 and 16 in d).

from people with early-stage Parkinson's disease. We normalized the vectors to zero mean and unit norm. Fig. 1f compares the performance GREEDI to the benchmarks with fixed $k = 50$ and varying number of partitions $m$. Similarly, Fig 1e shows the results for fixed $m = 10$ and varying $k$. We find that GREEDI significantly outperforms the benchmarks.

***Large scale experiments with Hadoop.*** Our second large scale experiment consists of 45,811,883 user visits from the Featured Tab of the Today Module on Yahoo! Front Page [26]. For each visit, both the user and each of the candidate articles are associated with a feature vector of dimension 6. Here, we used the normalized user features. Our experimental setup was a cluster of 5 quad-core machines running Hadoop with the number of reducers set to $m = 32$. Each reducer performed the lazy greedy algorithm on its own set of 1,431,621 vectors ($\approx$34MB) in order to extract 128 elements with the highest marginal gains w.r.t the local elements of the dataset in that particular partition. We then merged the results and performed another round of lazy greedy selection on the merged results to extract the final active set of size 128. The maximum running time per reduce task was 2.5 hours. Fig. 1g shows the performance of GREEDI compared to the benchmarks. We note again that GREEDI significantly outperforms the other distributed benchmarks and can scale well to very large datasets.

## 6   Conclusion

We have developed an efficient distributed protocol GREEDI, for maximizing a submodular function subject to cardinality constraints. We have theoretically analyzed the performance of our method and showed under certain natural conditions it performs very close to the centralized (albeit impractical in massive data sets) greedy solution. We have also demonstrated the effectiveness of our approach through extensive large scale experiments using Hadoop. We believe our results provide an important step towards solving submodular optimization problems in very large scale, real applications.

**Acknowledgments.**   This research was supported by SNF 200021-137971, DARPA MSEE FA8650-11-1-7156, ERC StG 307036, a Microsoft Faculty Fellowship, an ETH Fellowship, Scottish Informatics and Computer Science Alliance.

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
