[Supplementary Material]

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

## Footnotes

[1] Kumar, R., Moseley, B., Vassilvitskii, S., & Vattani, A. "Fast greedy algorithms in mapreduce and streaming." Proceedings of the 25th ACM Symposium on Parallelism in Algorithms and Architectures, ACM, 2013.

[2] Datasets are available at http://fimi.ua.ac.be/data/

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

# Appendix A: Proofs

This appendix presents the complete proofs of theorems presented in the article. For a set function $f$, we use the notation $f(S \mid S') = f(S \cup S') - f(S')$.

**Proof of Theorem 4.1**

$\Rightarrow$ direction:
The proof follows from the following lemmas.

**Lemma 6.1.** $\max_i f(A_i^c[k]) \geq \dfrac{1}{m} f(A^c[k])$.

**Proof:** Let $B_i$ be the elements in $V_i$ that are contained in the optimal solution, $B_i = A^c[k] \cap V_i$. Then we have:
$f(A^c[k]) = f(B_1 \cup \ldots \cup B_m) = f(B_1) + f(B_2|B_1) + \ldots + f(B_m|B_{m-1}, \ldots, B_1)$. Using submodularity of $f$, for each $i \in 1 \ldots m$, we have $f(B_i|B_{i-1} \ldots B_1) \leq f(B_i)$ and thus, $f(A^c[k]) \leq f(B_1) + \ldots + f(B_m)$. Since, $f(A_i^c[k]) \geq f(B_i)$, we have $f(A^c[k]) \leq f(A_1^c[k]) + \ldots + f(A_m^c[k])$. Therefore, $f(A^c[k]) \leq m \max_i f(A_i^c[k])$. $\qquad\square$

**Lemma 6.2.** $\max_i f(A_i^c[k]) \geq \dfrac{1}{k} f(A^c[k])$.

**Proof:** Let $f(A^c[k]) = f(\{u_1, \ldots u_k\})$. Using submodularity of $f$, we have $f(A^c[k]) \leq \sum_{i=1}^k f(u_i)$. Thus, $f(A^c[k]) \leq k f(u^*)$ where $u^* = \arg\max_i f(u_i)$. Suppose that the element with highest marginal gain ($u^*$) is in $V_j$. Then the maximum value of $f$ on $V_j$ would be greater or equal to the marginal gain of $u^*$, i.e. $f(A_j^c[k]) \geq f(u^*)$ and since $f(\max_i f(A_i^c[k])) \geq f(A_j^c[k])$, we can conclude that $f(\max_i f(A_i^c[k])) \geq f(u^*) \geq \frac{1}{k} f(A^c[k])$. $\qquad\square$

Since $f(A^d[m,k]) \geq \max_i f(A_i^c[k])$; from Lemma 6.1 and 6.2 we have $f(A^d[m,k]) \geq \frac{1}{min(m,k)} f(A^c[k])$.

$\Leftarrow$ direction:
Let us consider a set of unbiased and independent Bernoulli random variables $X_{i,j}$ for $i \in \{1, \ldots, m\}$ and $j \in \{1, \ldots, k\}$, i.e., $\Pr(X_{i,j} = 1) = \Pr(X_{i,j} = 0) = 1/2$ and $(X_{i,j} \perp X_{i',j'})$ if $i \neq i'$ or $j \neq j'$. Let us also define $Y_i = (X_{i,1}, \ldots, X_{i,k})$ for $i \in \{1, \ldots, m\}$. Now assume that $V_i = \{X_{i,1}, \ldots, X_{i,k}, Y_i\}$, $V = \bigcup_{i=1}^m V_i$ and $f_{S \subseteq V}(S) = H(S)$, where $H$ is the entropy of the subset $S$ of random variables. Note that $H$ is a monotone submodular function. It is easy to see that $A_i^c[k] = \{X_{i,1}, \ldots, X_{i,k}\}$ or $A_i^c[k] = Y_i$ as in both cases $H(A_i^c[k]) = k$. If we assume $A_i^c[k] = \{X_{i,1}, \ldots, X_{i,k}\}$, then $B = \{X_{i,j} | 1 \leq i \leq m, 1 \leq j \leq k\}$. Hence, by selecting at most $k$ elements from $B$, we have $H(A^d[m,k]) = k$. On the other hand, the set of $k$ elements that maximizes the entropy is $\{Y_1, \ldots, Y_m\}$. Note that $H(Y_i) = k$ and $Y_i \perp Y_j$ for $i \neq j$. Hence, $H(A^c) = k \cdot m$ if $m \geq k$ or otherwise $H(A^c[k]) = k^2$.

**Proof of Theorem 4.2**

Let us first mention a slight generalization of the performance guarantee for the standard greedy algorithm. It follows immediately from the argument in [10], see, e.g., [5].

**Lemma 6.3.** *Let $f$ be a non-negative submodular function, and let $A^{gc}[q]$ of cardinality $q$ be the greedy selected set by the standard greedy algorithm selecting $k$ elements. Then,*

$$f(A^{gc}[q]) \geq \left(1 - e^{-\frac{q}{k}}\right) f(A^c[k]).$$

By Lemma 6.3 we know that $f(A_i^{gc}[\kappa]) \geq (1 - \exp(-\kappa/k)) f(A_i^c[k])$. Now, let us define

$$B^{gc} = \cup_{i=1}^m A_i^{gc}[\kappa], \quad \tilde{A}[\kappa] = \arg \max_{S \subseteq B^{gc} \& |S| \leq \kappa} f(S).$$

Then by using Lemma 6.3 again, we obtain

$$f(A^{\text{gc}}[m,\kappa,l]) \geq (1 - \exp(-l/\kappa))f(\tilde{A}[\kappa]) \geq \frac{(1 - \exp(-l/\kappa))(1 - \exp(-\kappa/k))}{\min(m,k)}f(A^{\text{c}}[k]). \quad (5)$$

**Proof of Theorem 4.3**

First, we need the following lemma.

**Lemma 6.4.** *If for each $e_i \in A^{\text{c}}[k], |N_\alpha(e_i)| \geq km \log(k/\delta^{1/m})$, and if $V$ is partitioned into sets $V_1, V_2, \ldots V_m$, where each element is randomly assigned to one set with equal probabilities, then there is at least one partition with a subset $A_i^{\text{c}}[k]$ such that $|f(A^{\text{c}}[k]) - f(A_i^{\text{c}}[k])| \leq \lambda \alpha k$ with probability at least $(1 - \delta)$.*

**Proof:** By the hypothesis, the $\alpha$ neighborhood of each element in $A^{\text{c}}[k]$ contains at least $km \log(k/\delta^{1/m})$ elements. For each $e_i \in A^{\text{c}}[k]$, let us take a set of $m \log(k/\delta^{1/m})$ elements from its $\alpha$-neighborhood. These sets can be constructed to be mutually disjoint, since each $\alpha$-neighborhood contains $m \log(k/\delta^{1/m})$ elements. We wish to show that at least one of the $m$ partitions of $V$ contains elements from $\alpha$-neighborhoods of each element.

Each of the $m \log(k/\delta^{1/m})$ elements goes into a particular $V_j$ with a probability $1/m$. The probability that a particular $V_j$ does not contain an element $\alpha$-close to $e_i \in A^{\text{c}}[k]$ is $\frac{\delta^{1/m}}{k}$. The probability that $V_j$ does not contain elements $\alpha$-close to one or more of the $k$ elements is at most $\delta^{1/m}$ (by union bound). The probability that *each* $V_1, V_2, \ldots V_m$ does not contain elements from the $\alpha$-neighborhood of one or more of the $k$ elements is at most $\delta$. Thus, with high probability of at least $(1 - \delta)$, at least one of $V_1, V_2, \ldots V_m$ contains an $A_i^{\text{c}}[k]$ that is $\lambda \alpha k$-close to $A^{\text{c}}[k]$. □

By Lemma 6.4, for some $V_i$, $|f(A^{\text{c}}[k]) - f(A_i^{\text{c}}[k]|) \leq \lambda \alpha k$ with the given probability. And $f(A_i^{gd}[\kappa]) \geq (1 - e^{-\kappa/k})f(A_i^{\text{c}}[k])$ Lemma 6.3. Therefore, the result follows using arguments analogous to the proof of Theorem 4.2.

**Proof of Theorem 4.4**

The following lemma says that in a sample drawn from distribution over an infinite dataset, a sufficiently large sample size guarantees a dense neighborhood near each element of $A^{\text{c}}[k]$ when the elements are from representative regions of the data.

**Lemma 6.5.** *A number of elements: $n \geq \dfrac{8km \log(k/\delta^{1/m})}{\beta g(\alpha)}$, where $\alpha \leq \alpha^*$, suffices to have at least $4km \log(k/\delta^{1/m})$ elements in the $\alpha$-neighborhood of each $e_i \in A^{\text{c}}[k]$ with probability at least $(1 - \delta)$, for small values of $\delta$.*

**Proof:** The expected number of $\alpha$-neighbors of an $e_i \in A^{\text{c}}[k]$, is $E[|N_\alpha(e_i)|] \geq 8km \log(k/\delta^{1/m})$. We now show that in a random set of samples, at least a half of this number of neighbors is realized with high probability near each element of $A^{\text{c}}[k]$.

This follows from a Chernoff bound:

$$P[|N_\alpha(e_i)| \leq 4km \log(k/\delta^{1/m})] \leq e^{-km \log(k/\delta^{1/m})} \leq (\delta^{1/m}/k)^{km}.$$

Therefore, the probability that some $e_i \in A^{\text{c}}[k]$ does not have a suitable sized neighborhood is at most $k(\delta^{1/m}/k)^{km}$. For $\delta \leq 1/k$, $k\delta^{km} \leq \delta^m$. Therefore, with probability at least $(1 - \delta)$, the $\alpha$-neighborhood of each element $e_i \in A^{\text{c}}[k]$ contains at least $4km \log(1/\delta)$ elements. □

**Lemma 6.6.** *For $n \geq \dfrac{8km \log(k/\delta^{1/m})}{\beta g(\frac{\varepsilon}{\lambda k})}$, where $\frac{\varepsilon}{\lambda k} \leq \alpha^*$, if $V$ is partitioned into sets $V_1, V_2, \ldots V_m$, where each element is randomly assigned to one set with equal probabilities, then for sufficiently small values of $\delta$, there is at least one partition with a subset $A_i^{\text{c}}[k]$ such that $|f(A^{\text{c}}[k]) - f(A_i^{\text{c}}[k])| \leq \varepsilon$ with probability at least $(1 - \delta)$.*

**Proof:** Follows directly by combining lemma 6.5 and lemma 6.4. The probability that some element does not have a sufficiently dense $\varepsilon/\lambda k$-neighborhood with $km\log(2k/\delta^{1/m})$ elements is at most $(\delta/2)$ for sufficiently small delta, and the probability that some partition does not contain elements from the one or more of the dense neighborhoods is at most $(\delta/2)$. Therefore, the result holds with probability at least $(1 - \delta)$. $\square$

By lemma 6.6, there is at least one $V_i$ such that $|f(A^c[k]) - f(A_i^c[k])| \leq \varepsilon$ with the given probability. And $f(A_i^{gd}[\kappa]) \geq (1 - e^{-\kappa/k})f(A_i^c[k])$ using Lemma 6.3. The result follows using arguments analogous to the proof of Theorem 4.2.

**Proof of Theorem 4.5**

Note that each machine has on the average $n/m$ elements. Let us define $\Pi_i$ the event that $n/2m < |V_i| < 2n/m$. Then based on the Chernoff bound we know that $\Pr(\neg\Pi_i) \leq 2\exp(-n/8m)$. Let us also define $\xi_i(S)$ the event that $|f_{V_i}(S) - f(S)| < \epsilon$, for some fixed $\epsilon < 1$ and a fixed set $S$ with $|S| \leq k$. Note that $\xi_i(S)$ denotes the event that the empirical mean is close to the true mean. Based on the Hoeffding inequality (without replacement) we have $\Pr(\neq \xi_i S| \leq 2\exp(-2n\epsilon^2/m)$. Hence,

$$\Pr(\xi_i(S) \wedge \Pi_i) \geq 1 - 2\exp(-2n\epsilon^2/m) - 2\exp(-n/8m).$$

Let $\xi_i$ be an event that $|f_{V_i}(S) - f(S)| < \epsilon$, for any $S$ such that $|S| \leq \kappa$. Note that there are at most $n^\kappa$ sets of size at most $\kappa$. Hence,

$$\Pr(\xi_i \wedge \Pi_i) \geq 1 - 2n^\kappa(\exp(-2n\epsilon^2/m) - \exp(-n/8m)).$$

As a result, for $\epsilon < 1/4$ we have

$$\Pr(\xi_i \wedge \Pi_i) \geq 1 - 4n^\kappa\exp(-2n\epsilon^2/m).$$

Since there are $m$ machines, by the union bound we can conclude that

$$\Pr((\xi_i \wedge \Pi_i) \text{ on all machines}) \geq 1 - 4mn^\kappa\exp(-2n\epsilon^2/m).$$

The above calculation implies that we need to choose $\delta \geq 4mn^\kappa\exp(-2n\epsilon^2/m)$. Let $n_0$ be chosen in a way that for any $n \geq n_0$ we have $\ln(n)/n \leq \epsilon^2/(mk)$. Then, we need to choose $n$ as follows:

$$n = \max\left(n_0, \frac{m\log(\delta/4m)}{\epsilon^2}\right).$$

Hence for the above choice of $n$, there is at least one $V_i$ such that $|f(A^c[k]) - f(A_i^c[\kappa])| \leq \varepsilon$ with probability $1 - \delta$. Hence the solution is $\epsilon$ away from the optimum solution with probability $1 - \delta$. Now if we confine the evaluation of $f(A_i^c)$ to data points only in machine $i$ then under the assumption of Theorem 4.4 we loose another $\epsilon$. Formally, the result at this point simply follows by combining Theorem 4.2 and Theorem 4.4.

## Appendix B: Additional Experiments

**Finding maximum cuts.** We also applied GREEDI to the problem of finding maximum cuts in graphs. In our setting we used a *Facebook-like social network* [27]. This dataset includes the users that have sent or received at least one message in an online student community at University of California, Irvine and consists of 1,899 users and 20,296 directed ties. Fig. 2 shows the performance of GREEDI applied to the cut function on graphs. We evaluated the objective function locally on each partition. Thus, the links between the partitions are disconnected.

This experiment violates several assumptions we made: 1) the cut function is submodular but not monotonic (and hence neither our theory holds, nor is the greedy algorithm guarantees to provide good solutions). 2) the cut function does not decompose additively over individual data points. Perhaps surprisingly, GREEDI still performs very well, and significantly outperforms the benchmarks. This suggests that our approach is quite robust, and may be more generally applicable.

Figure 2: Facebook-like social network

Figure 3: Mean and standard deviation of the ratio of distributed to centralized solution for budget $k = 20$ with varying number of machines $m$ on a *Facebook-like social network*.

**Comparision with** GREEDY SCALING. Kumar et al.[1] recently proposed an alternative approach – GREEDYSCALING – for parallel maximization of submodular functions. GREEDYSCALING is a randomized algorithm that carries out a number (typically less than $k$) of MapReduce computations. This work was not available at the time of submission of our original manuscript. In the following, we briefly discuss some of the main differences.

First, GREEDYSCALING assumes that the objective function can be evaluated on each machine for any given set. In many realistic scenarios however, the objective function may depend on the entire dataset and different machines may not have access to the full dataset. We explicitly addressed this issue in Section 4.5.

Second, Kumar et al. describe conditions under which GREEDYSCALING achieves close-to-centralized performance. These conditions do not require geometric structure, as we do in our analysis. In contrast to our results, however, it is required that the ratio between the largest and smallest marginal gains of $f$ is bounded by $\delta$, which restricts generality (e.g., for the entropy function, $\delta$ can be exponentially small in $n$).

Last, while our approach only requires two rounds of MapReduce GREEDYSCALING, the number of rounds depend on the quantity $\delta$ which may be unbounded.

We applied GREEDI to the submodular coverage problem in which given a collection $S$ of sets, we would like to pick at most $k$ sets from $S$ in order to maximize the size of their union. We compared the performance of our GREEDI algorithm to the reported performance of GREEDYSCALING on the same datasets (*Accidents* and *Kosarak* [2]) used by Kumar et al. As Fig 4 shows, GREEDI significantly outperforms GREEDYSCALING on the "*Accidents*" dataset and its performance is comparable to that of GREEDYSCALING in the "*Kosarak*" dataset.

Figure 4: Performance of GREEDI compared to GREEDYSCALING. a) and b) show the ratio of distributed to centralized solution on *Accidents* and *Kosarak* datasets with 340,183 and 990,002 elements, respectively. The results are reported for varying budget $k$ and varying number of machines $m = n/\mu$ where $\mu = O(kn^\delta \log n)$ and $n$ is the size of the dataset. The results are reported for $\delta = 1/2$.