[Reviews · NeurIPS 2013]

Submitted by Assigned_Reviewer_4

This paper describes a two-stage approach to distributed submodular maximization, known as GreeDi. Error bounds are derived for the GreeDi algorithm, which provide a theoretical guarantee on the greedy approximate solution to the centralized solution. The authors demonstrated the effectiveness on 6 data sets.

Here are my comments.
1. It is confusing to have kappa and k in Theorem 4.2. It is hard to distinguish them in reading.
2. Regarding the bound in Theorem 4.2, it would be helpful to comment on the tightness. I note that there is a factor min(m,k) inside.
3. In experiments, it would be informative to report generalized performance, such as negative log predictive probability. The decrease on objective functional is expected, while it is interesting to know how much it affect generalization.
4. It is unclear which experiments are handling decomposable functions in Section 5.
5. In Figure 1(e), the label on x axis should be k.
6. In Figure 1(f), why the ratio at the smallest m starts below 1, while it starts from 1 in Figure 1(a)-1(d).
7. How do we explain the dip when k=10 in Figure 1(c)?
8. Adding references to Today Module of Yahoo!, that helps readers carry out related research.
Summary: It is an interesting study for distributed data mining.

Submitted by Assigned_Reviewer_6

I think this is a really nice paper. It's addressing an important problem area, gives a simple practical algorithm that's easily implemented, and the empirical results are good. The theoretical analysis is well done and honest (this approach won't always work well, but the assumptions that need to be made are reasonable for learning tasks).

My main criticism of the paper is that it feels squeezed for space. The experimental section, in particular, is much too terse around explanations of baseline comparison methods and explication of the results. (Figure 1 has a ton going on in it.) I find the strong performance of greedy/max interesting in itself any maybe worth a little discussion. Also, there's no discussion of run-time or cpu cost for any of the methods -- odd for a paper that's pushing on scalability, and should definitely be addressed.

I'm surprised that the idea of lazy evaluation wasn't discussed more in this paper. This seems to give huge wins for efficiency, and should certainly be mentioned as a practical trick for a scalability paper, since some of the folks looking at this may be more engineering-oriented and not know about this. I also wonder if lazy-evaluation + mapreduce is a useful way to get by without the approximation method of this paper -- if you can eat the cost of the first round to select the first element, subsequent rounds will be really very cheap. (You can do things like "evaluate next 100 from the list" in parallel; if you hit on #55 on the list you've wasted some computation but are still doing fine compared to a full evaluation of all candidates). The first paragraph of 3.2 suggests that this is impractical for large k, but for large k things are expensive anyway.

The phrase "diminishing returns" should be mentioned in the introductory part of section 3 (or in the introduction). I feel this is the most intuitive way to understand submodularity, for those who are not familiar with it.

In the paper, it's not clear what's meant by "fit on one machine". Is the issue what can fit in RAM? On disk? The amount of CPU processing available? The first paragraph of section 2 makes it seem like CPU cost is the main bottleneck, but often times disk i/o is equally large an issue. What benefits (if any) can we get from a large-RAM machine with 16 or more cores?

Since I'm asking for more detail in some places, I need to propose some things to cut:
-- I'd definitely cut section 6 -- it doesn't add anything to the paper.
-- I think we can also live without pseudo-code for algorithm 1.
-- The first two sentences of paragraph 3 of section 1, and the last sentence of this paragraph, can be cut. The remaining sentence can be merged with the one above it.
-- The first paragraph of section 2 can be significantly shortened.
-- Paragraphs 2 and 3 of section 3.2 can be cut. (You might also mention the naive approach of sampling the data set down to a size that can fit on one machine and run the standard algorithm here.)


Summary: The paper proposes a simple but novel method for submodular maximization in a distributed framework. They show that it is a sound approach with theoretical analysis under some reasonable assumptions, and report good results across several possible applications.

Submitted by Assigned_Reviewer_7

This paper introduces the GreeDi algorithm to maximize monotone submodular functions subject to a cardinality constraint in a distributed system environment, in particular mapreduce. The authors experiment against an exhaustive array of datasets and also prove that the distributing of work across many machines maintains the objective to within a reasonable bound of a centralized algorithm.

The function optimized must be "decomposable" i.e. composed of the sum of many submodular functions, so that the function does not depend on the entire dataset.

I would like to see the following questions explicitly answered:

1) Exactly How much communication is required between the mappers and reducers in the mapreduce implementation? i.e. how much data needs to be shuffled? this is the communication cost in this setup.

2) Exactly how many items could be reduced to a single key? This measures how overloaded a single machine may become.

3) How many iterations needed in the worst case?
Summary: This paper introduces the GreeDi algorithm to maximize monotone submodular functions subject to a cardinality constraint in a distributed system environment, in particular mapreduce. The authors experiment against an exhaustive array of datasets and also prove that the distributing of work across many machines maintains the objective to within a reasonable bound of a centralized algorithm.

Submitted by Assigned_Reviewer_8

This paper provides a novel algorithm for large scale submodular function maximization. The main contribution is a distributed algorithm, using MapReduce, where the ground set is partitioned and each machine operates on a subsets of the ground set. These solutions are merged, to find an approximate subset. They provide approximation guarantees for their approach and also show that it is tight. They also investigate the performance of their algorithm under specific assumptions on the datasets or the function.

I think overall the authors try to solve a very challenging problem, which could have a lot of utility in large scale real world applications. I also feel that the experimental validation is thorough and extensive. I also appreciate that this problem is very challenging and it would really hard to obtain satisfactory performance guarantees without additional assumptions.

I was somewhat disappointed with the theoretical analysis. It seems that the guarantees are pretty weak; I acknowledge that the worst case analysis shows that the algorithm is tight. However, this is expected particularly for a very bad choice of partitions V1, V2, ... I was expecting some kind of dependence on the choice of the the distributions, or even a heuristic of what choices of distributions might work. The main guarantee (Theorem 5.1) seems almost a linear factor in m and k, which is somewhat discouraging. As the proof technique reveals, and even otherwise, it is easy to see that a simple modular upper bound based approximation gives a factor k approximation. Given this, it is not immediately clear how GREEDI performs theoretically w.r.t a simple modular upper bound particularly for large m (which is of practical relevance), though I am sure the modular upper bound based algorithm will perform very badly in practice.

There is one extremely important aspect, however, which is very loosely described in the paper. I think this should be clarified much better. A number of practical submodular functions are graph based submodular functions (this includes, for example, the graph cut like function and the exemplar based clustering objective from 3.1 -- A small clarification here is this objective is essentially a form of facility-location objective with a similarity instead of a distance function) In these cases, evaluating the function requires a sum over all the elements in V, even though the set under which the function needs to be evaluated is considerably smaller. In these cases, it is not clear how to evaluate the function. More specifically, a main motivation of this approach (lines 122-127) is that the datasets are extremely large and would possibly not fit within any single machine. However each of the individual machines would still need to compute the outer sum over V in a graph based objective (say the facility location objective). Under these circumstances, it is not clear how to run the algorithm. Possibly, there could be a time reduction due to the distribution, but it is not clear how this will help in terms of memory. One solution could be to evaluate the function just on the subsets Vi (i.e the outer sum may only be over the Vi's), but this would then change the submodular function and the guarantees would no longer hold. Overall, I think this is a major issue which should be clarified in the rebuttal.

Some minor suggestions are that the proof of theorem 4.1 (specifically the tight instance) is very hard to grasp. Maybe that can be better explained? Also, I think over smaller datasets the actual greedy algorithm should be compared to GREEDI (just to see how the performance is with respect to the best serial algorithm). I would also love to see a timing analysis of the algorithms and memory requirements etc. in the experimental results.
Summary: I think the paper addresses a novel (and challenging) problem of distributed techniques for submodular maximization. This algorithm could have a lot of practical impact. However, the theoretical contribution of this paper is weak.

Submitted by Assigned_Reviewer_9

This paper is on solving submodular maximization problems at scale.
In particular, the paper looks at the classical greedy algorithm for
submodular maximization under cardinality constraints and offers
modifications to run these algorithms on massive data.

The problem itself is quite motivated. There have been a few earlier
work on trying to "speed up" or "parallelize" the inherently
sequential greedy algorithm for the submodular maximization problem.
MapReduce as a programming paradigm to express the algorithm is also
well motivated.

The main technical contribution of the paper is an analysis of the
following two-round algorithm: the input is split across the machines
and each machine approximates the solution to its part of the input
(by running a sequential greedy algorithm) and then the individual
solutions are combined to obtain the final algorithm. The key point
here is for each machine to output a solution of size more than k/m (k
= desired solution size, m = number of machines). The analysis itself
is quite simple and the paper shows inherent dependence on both k and
m. The paper also has sundry results for special cases, for eg,
smooth spaces and for decomposable functions.

The experimental results are reasonable showing the efficacy of the
two-round algorithm when compared to standard greedy.

On the positive side, the paper addresses an important problem and
proposes a practical modification of the standard algorithm.

The main negatives of the paper are the following:

1. The paper is near-trivial on the theory front. The analysis is so
obvious from a theoretical perspective. Some of the proofs are
repetitive in nature and the seemingly strong results stem from quite
strong assumptions. Randomized input partition is not taken advantage
of in Theorem 4.2 (or show a lower bound).

2. There is no round-memory-approximation tradeoff. The paper is
restrictive in its results and the approach itself is unclear to be
generalized to multiple rounds. In this sense, it is significantly
weaker than the earlier work (WWW paper of Chierichetti et al or the
SPAA paper of Lattanzi et al).

3. The experimental results contain several needless baselines
(random/random, for example). The authors do not try to bring out the
importance of oversampling by modifying the greedy/merge to make
greedy as a function of the size of the local solutions.

Additional comments:

1. The paper should investigate if slightly stronger bounds can be
proved for Theorem 4.2 when the inputs are randomly partitioned.

2. The authors may want to look at the SPAA 2013 paper of Kumar,
Moseley, Vassilvitskii, and Vattani that addresses a similar problem
but in more general context and provides a multi-round and better
approximation algorithm.

3. It might be possible to "merge" the results in Theorem 4.3 and 4.4
since they seem to be using related assumptions (neighborhood size for
metric spaces vs. growth function of a metric, which is the volume).

4. The authors may want to compare their algorithm with the Chierichetti
et al paper and the SPAA 2013 paper.

5. page 4, line 167: "explain what is "suitable choice""

6. page 5, line 227: didnt get the comment about "... unless P = NP". why does it follow?

Summary: A theoretically weak paper addressing an important problem.
Author Feedback

Author rebuttal: We kindly thank the anonymous reviewers for their careful review.

Reviewer 4
- Thm 4.1 & 4.2:
The bound is tight in the worst case analysis for the two-rounds scheme we proposed and is linear in min(m,n). We agree that it would be interesting to analyze the effect of randomized input partitions for general submodular functions. However, for practical applications we find it very important to exploit additional structure (geometric, high density, decomposability) beyond just submodularity for which we obtain much stronger theoretical guarantees. We also reported experimental results that show the ratio between the distributed and centralized algorithm is close to one.

- Which experiments are handling decomposable functions?
Exemplar based clustering.

- In Figure 1(f), why the ratio at the smallest m starts below 1.
The cut function is submodular but not monotone. Hence, any non-empty cut among k elements of the first greedy round in GreeDi always have less than k elements. Therefore, for m=1, we won't obtain the same result for the distributed and centralized solutions.

- How do we explain the dip when k=10 in Figure 1(c)?
For k = 1, the greedy algorithm returns the optimal solution. For any k>1 (e.g., k=10), it’s a general effect in all experiments. For large k, we usually recover solutions very close to the centralized one.

Reviewer 6
- Discussion about run-time and memory requirements:
We explicitly mentioned the costs for the Hadoop implementation. In general, the memory requirement is only n/m.

- Discussion about lazy evaluation (+ Mapreduce):
As discussed in section 3.2, for large k -which is the case of our attention- this approach requires a high amount of communication, which is impractical for MapReduce style computations. In all of our experiments, we used lazy evaluation on each machine.

- Meaning of "fit on one machine":
It is not possible to load a dataset of hundreds of gigabytes in memory. Even if we can load and calculate the marginal gains on one machine with multiple cores, merging and sorting the result takes a huge time.

Reviewer 7
- Amount of communication between mappers and reducers:
In the first round no communication takes place and each machine performs its task locally. In the second round, the k results of the m machines are merged, i.e., k*m data points, and greedy selection will be performed on the merged results.

- How many items could be reduced to a single key?
In the first round the data is partitioned equally among m machines, hence n/m items have the same key. In the second round all the k*m elements are merged to the same key.

- How many iterations needed in the worst case?
GreeDi has only 2 rounds of MapReduce.

Reviewer 8
- Theoretical guarantee in the worst case:
Please see the answer to Reviewer 4.

- GREEDI w.r.t. a simple modular upper bound.
Optimizing a simple modular upper bound would give a O(k) approximation, not min(m,k). In some important practical settings k can be much bigger than m. As stated above, for many realistic objective functions, we obtain much better approximations (both in theory and experiments).

- Local evaluations of submodular functions.
We fully agree with the reviewer that in many practical settings, evaluating f exactly requires access to the full data set. For this precise reason we study functions with decomposable structure in Section 4.5. along with the performance guarantee of GreeDi in such settings. Moreover, Fig 1(f) (finding max cut) suggests that our approach is quite robust, and may be more generally applicable.

- GREEDI versus centralized greedy alg.
In all of the experiments (except the very large Y! webscope dataset), we plotted the ratio of the distributed to the centralized greedy algorithm. They are indeed very close, suggesting GreeDI provides very good approximations.

Reviewer 9
- Round-memory-approximation tradeoff:
One of the advantages of our approach is its simplicity, i.e., it can be performed in only two rounds while still providing a constant approximation guarantee. It doesn’t require to be implemented in multiple rounds as the works mentioned by the reviewer, which we argue is a benefit in practice. GREEDI also provides highly accurate approximations in practical settings.

- Baselines in the experimental results:
We believe that all the baselines show interesting trends and are very informative. As shown in Fig 1. greedy/max, greedy/merge, random/greedy perform closely to GreeDi. It is even surprising to see that random/random outperforms greedy/merge in Y! webscope in Fig. 1(e).

- SPAA 2013:
This is a contemporary work which was not available at the time of the NIPS submission. In the following we briefly discuss some differences.

In SPAA 2013, it is assumed that the objective function can be evaluated for any given set. In many realistic scenarios the objective function may depend on the entire data set and different machines may not have access to the entire dataset. We explicitly addressed this issue in Section 4.5.

In SPAA 2013, it is assumed that the ratio between the largest and smallest marginal gain of f is bounded by Delta, which restricts generality (e.g., for the entropy function, Delta can be exponentially large in n).

In contrast, we identified natural structures that are realistic in large-scale settings for which we obtained strong theoretical guarantees.

Our approach is a two-round algorithm that can be easily implemented in MapReduce. In contrast, in SPAA 2013, the number of rounds depend on Delta which may be unbounded.

-Explain what is "suitable choice”:
It’s common to consider either origin or mean value of the dataset as the auxiliary element e_0 (ref [4]).

- "unless P = NP". why does it follow?
Maximizing a monotone submodular function with cardinality constraint is NP-hard.